# Social Housing Life Cycle Management: Workflow for the Enhancement of Digital Management Based on Building Information Modelling (BIM)

**Manuel Castellano-Román [1] , Antonio Garcia-Martinez [2],* and María Luisa Pérez López [3]**

1 Research Group HUM 799, Heritage Knowledge Strategies, Department of Architectural Graphic Expression, University Institute of Architecture and Construction Sciences IUACC, Universidad de Sevilla, 41012 Seville, Spain; manuelcr@us.es
2 Research Group TEP 130, Architecture, Heritage and Sustainability: Acoustics, Lighting, Optics and Energy Department of Architectural Construction, Research, University Institute of Architecture and Construction Sciences IUACC, Universidad de Sevilla, 41012 Seville, Spain
3 Andalusian Housing and Rehabilitation Agency AVRA, Ministry of Development, Infrastructures and Spatial Planning, Junta de Andalucía, 41018 Sevilla, Spain; marial.perez.lopez@juntadeandalucia.es
* Correspondence: agarcia6@us.es; Tel.: +34-639-46-56-69

**Abstract:** The management of the life cycle of large publicly owned social housing complexes requires a large amount of human and technological resources, the optimization of which is a desirable and shared objective. This article proposes a workflow for the enhancement of these management processes based on BIM (Building Information Modelling), a methodology capable of integrating architectural information into a three-dimensional graphic model. The proposed workflow defines the basic characteristics of the BIM model oriented toward sustainable building management and its relationship with the key moments of its life cycle. It also analyzes the architectural information associated with the models and determines which parameters are optimal for their completion from the BIM models in terms of reliability, auditability, and automation. For this purpose, a case study has been developed for a multifamily residential building in Malaga (Spain), owned by the Andalusian Housing and Rehabilitation Agency AVRA, a public agency that manages a housing stock of more than 70,000 dwellings.

**Keywords:** social housing management; building information modelling (BIM); computerised maintenance management system (CMMS)

## 1. Introduction

The economic, social, and environmental sustainability of public and private buildings depends, to a large extent, on efficient and effective maintenance and conservation management during their useful life cycle. To this end, numerous IT tools have been designed in recent decades, from simple databases to specialized software known as Computerized Maintenance Management Systems (CMMS) [1]. These tools provide information on the key points of buildings managed by an organization, supporting the making of informed decisions regarding them [2].

To complement these digital resources for management, some authors have proposed the convenience of incorporating Building Information Modelling (BIM), making use of its capacity to integrate architectural information into a three-dimensional graphic model [3–5]. The advantages that stand out are the three-dimensional visual presentation of the model, its ability to generate effective communication between the agents involved (e.g., technicians, users, managers, and politicians), assistance in the control of maintenance (e.g., refurbishments, repair operations, and its costs,) and its usefulness as a bridge to other technological applications [6–15]. In parallel to the development of BIM, the digital maintenance and conservation phase has experienced significant progress [16–24].

In the specific context of the application of BIM to the management of social housing, references are scarcer. In Europe, a case study has been carried out in Triolo, France, which proposes a social housing management strategy based on the generation of a BIM model and the direct management of it [25]. It characterises the advantages of using BIM but has the limitations of lacking external management software and not addressing the issue of managing many buildings.

Another European study, located in the United Kingdom, provides experience of the use of BIM in the refurbishment process of a social housing building. This experience focuses on the time of the intervention and not so much on the subsequent management, highlighting the advantages of the fourth dimension of BIM, i.e., the time sequence in the models [26].

In Oceania, a New Zealand experience promoted by a public institution, the Wellington City Council, is of interest. This experience focuses on the benefits that BIM methodology can offer a public manager in predicting maintenance costs and planning management to extend the useful life of buildings. BIM models act as a data collector whose further management is in the process of development [27].

In the Americas, an experience developed in Chile has been published in which BIM is introduced to manage the moment when newly constructed social housing buildings are occupied by their inhabitants. It shares with the experience presented in this article attention to the phase of the useful life of the building but focuses on the characterization of damages in the building, recorded in the model itself and not in their management from external applications [28]. Finally, an experience from Brazil provides the perspective of client requirements in social housing projects using BIM, focusing on the need to adjust the information structure of the models to the objectives of the latter [29].

### 1.1. Problem Statement

Despite these advances, organizations that have to manage a large number of social housing estates find it very difficult to apply these technologies. The main reason for this is that the buildings they manage are, for the most part, buildings of a certain age, built before the widespread use of digital graphics and information tools. First, that implies major problems with alphanumeric document management, generally solved with basic database tools, and, to an even greater extent, problems of graphic information management, which is stored on paper, raster images from the digitization of paper plans, or, in the best of cases, on vector CAD support. Second, this alphanumeric and graphical information must be converted into an effective real estate decision-making assistant. To do so, the data must be reliable and auditable. Furthermore, if data correction and updating is necessary, the process should be as automated as possible.

This is the case for the Andalusian Housing and Rehabilitation Agency (AVRA), one of the public organizations with the largest number of dwellings managed, with 73,989 houses that provide accommodation to more than 300,000 people, on whose experience the case study will be based [30].

AVRA has wide and varied real state under its supervision, the management of which involves a major documentation problem, including graphics, which is often present in preventive conservation projects [31]. Generally, this documentation is on paper, the result of the predigital tradition of managing records through copies and reproducible means, typical of the last century. On other occasions, especially in the 1990s, paper deliveries were accompanied by copies on CD media of planimetry developed in CAD in a digital environment. The latter generally contain dwg and dxf files, whereby the information is systematized in folders, layers, and linked files, from which a printed dump can be obtained on paper or in PDF format. Despite this, dossiers have continued to appear in print well into the current century, either on paper or in PDF format, a format that was officially launched as an open standard in 2008 and published by the International Organization for Standardization (ISO) as ISO 32000-1.

In the General Archive of AVRA, this agency has only commissioned paper projects since its foundation in 2013. However, the public housing stock is made up of social housing that comes from various entities (ministries, councils, and municipal companies), most of which is from the second half of the last century, which are graphically recorded on paper and deposited in the Provincial Offices, not in the General Archive. Project copies on CD began to be ordered at the beginning of this century but have not been inventoried. This last medium usually presents the problem of difficult long-term conservation since it is highly vulnerable and contains information that is difficult to retrieve. In the case of the rehabilitation of historical buildings, or those built prior to the middle of the twentieth century, which represent a minority in relation to the total computation of the agency's social housing, there is a definite lack of updated documentation, although on rare occasions documentation is available in provincial or municipal historical archives.

Currently, AVRA uses software for the management of its social housing, named herein as CMMS-AVRA. It organizes a database that contains essential information on each managed building. The content and structure of this database are established in the Law 8/2013, of 26 June, on urban rehabilitation, regeneration, and renovation (BOE, 27 June 2013). This law describes the information required in a Building Assessment Report, a document standardized by the Spanish Government [32] that records the basic architectural characteristics and state of conservation of buildings. Based on this information, AVRA plans the rehabilitation actions to be carried out in each campaign.

*1.2. Objectives*

The main objective of this article is to show how the management of the digital lifecycle of public social housing parks can be improved with the use of the BIM methodology. To achieve this main objective, the following specific objectives are designed. The first is the proposal of a strategy to be managed for the graphic documentation of the buildings, capable of providing the necessary information to set up the BIM model.

The second specific objective is the analysis of the information structure of the building managers' own applications, in this case the CMSS-AVRA software, to select the key parameters whose determination can be improved from the BIM model. The aim is to select those that the BIM model can complete with guarantees of reliability, auditability, and automation.

The third specific objective is the characterization of the BIM model oriented to the sustainable management of its life cycle. This characterization will be determined in terms of a modelling strategy and linkage with the different situations in which the building may find itself during its life cycle.

The fourth specific objective is to resolve the connection of the BIM model with the building managers' own applications, in this case CMSS-AVRA software, enabling the flow of information between the two.

## 2. Methods

*2.1. Current Workflow Analysis*

Information on the current situation of the workflows carried out by AVRA in the management of its social housing was extracted from the performance of structured face-to-face and online interviews. These interviews were carried out by the research team in the period January-February 2021. The interviewees were the technicians responsible for the maintenance of the AVRA buildings. These interviews focused on the current common process that the agency is carrying out in the management of its social housing and the processes that could be optimized using the BIM methodology.

The information required for the analysis and evaluation of the improvement possibilities was obtained from unstructured interviews with various building managers (IT experts, architects, engineers, and managers). In these interviews, an attempt was made to determine the real needs of AVRA in relation to maintenance and the real possibilities of

implementing BIM-based conservation management, considering the knowledge of these resources held by the technical personnel in charge of building conservation management.

The final evaluation of this process is presented in Section 1.1 as a problem statement. Although a management software is in place, CMSS-AVRA, the management model is highly analogue. The process of data entry and maintenance of the management system is essentially manual, and updated graphic documentation is rarely available. By not benefitting from the automation provided by methodologies such as BIM, maintenance is costly and inefficient, according to the managers themselves.

### 2.2. Case Study Selection

To achieve the objectives, one of the multifamily residential buildings managed by AVRA was taken as a case study (Figure 1). This building, built in Malaga, is representative of the type of property that AVRA manages, and lies both in the field of BIM management of social housing and in the generation of information models of existing buildings.

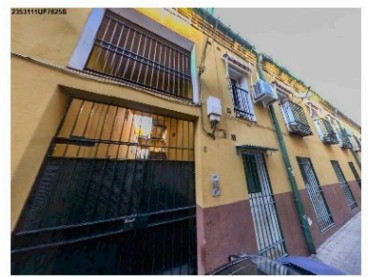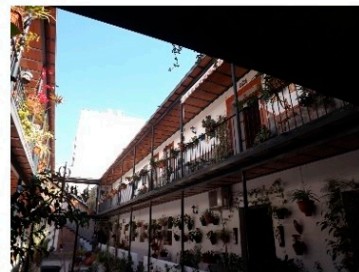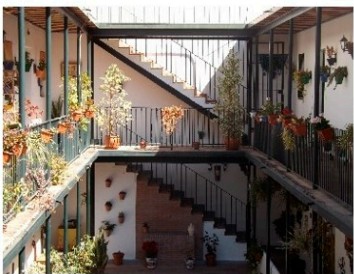

**Figure 1.** Case study, building on Lemus Street nº5 (Malaga). General Directorate of Cadaster and authors.

This building was constructed in 1987 and is located between Lemus Street and Plaza Bravo, in the Trinidad/Perchel neighborhood of Malaga. Its floor plan is rectangular, with two-storey pendent blocks connected through the interior patio and the first-floor gallery, evoking the image of an old neighborhood corral that previously existed at that location. It has two façades, Lemus Street and Plaza Bravo, from which the building is accessed. All the social housing in the promotion remains under a social rental scheme.

### 2.3. Digital Capture

The generation of the BIM model is based on the 2D CAD survey of the building, supplied by AVRA. In order to verify the degree of approximation of the model with the physical reality of the property and to evaluate the margin of error that the utilization of previous standard documentation supposes, two data captures were also made using digital technology: (a) a static laser scanner, based on taking data through static scanning locations; and (b) a hand-held scanner that performs data collection dynamically, accompanying the movement of a person moving through the property [33]. The equipment used was, firstly, a LEICA BLK360 IMAGING LASER SCANNER, capable of capturing 360,000 points per second, with a range of 60 m and a precision of 4 mm; it also includes thermal, laser, and visible light images, and is able to create a 360° scan in just 3 min. Secondly, a ZEB-REVO hand-held scanner was used, with a maximum range of 30 m, 43,000 points per second, and a relative precision of 1 to 3 cm.

The work carried out with the LEICA BLK360 IMAGING LASER SCANNER consisted of collecting data through 21 scanning locations in the common spaces of the building, starting from the centre of the courtyard. The interior of the dwellings was avoided since it would have required a scanning location for each room and would therefore have considerably increased the time of field work.

Through the LEICA Data Manager program, the download of the 21 BLK360 files took 60 min and their importing through the LEICA Cyclone Register program took another 60 min, while the recording of the 21 positions for their unification and adjustment took

120 min, and their exporting in rcp format, occupying 18 Gb of information, took another 210 min. A total of 555 min was spent, during which a massive data collection of points was obtained, including the management of the chromatic information of each point. This allowed dynamic images of optimal resolution to be obtained that provide a virtual tour of the building.

The metric capture using the ZEB-REVO hand scanner was carried out in two successive data collections. The first collection included the interior of one of the houses, the entire gallery, stairs, and a roof terrace. In the second data collection, the exterior façades were scanned (Figure 2). The starting point for each data collection was the same, with a duration of each moving image of 15 min.

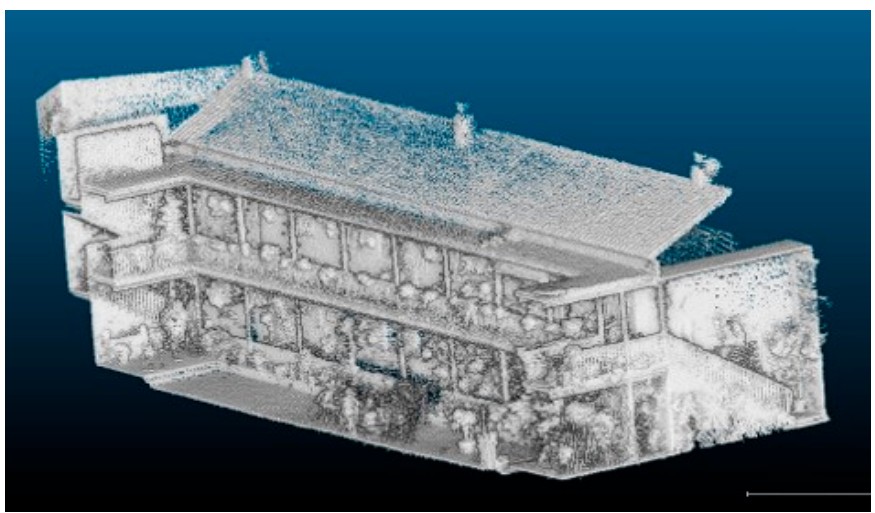

**Figure 2.** Raw point cloud obtained from the scan: sectioned axonometry of the point cloud.

The clerical work consisted of dumping the point clouds from the scanners into universal format files, with an e57 extension from the scanner rental company, and of sending them by email link, which took approximately 10 min. Our team then imported these files into Autodesk Recap, to convert them into an Autodesk rcp file, compatible with Autodesk AutoCAD and Revit programs (Figure 3), which took another 10 min.

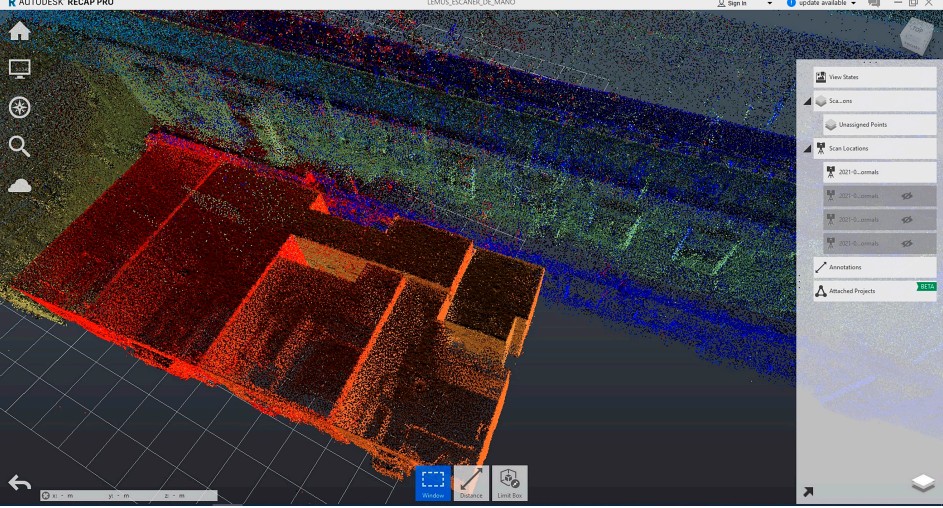

**Figure 3.** Point cloud after importing into Autodesk Recap.

This point cloud is scaled and oriented vertically from the origin, hence it is only necessary to orient it horizontally and move it based on a known georeferenced point. In our case, this cloud was transferred to the model that had already been prepared from the existing planimetry to verify the possible readjustments to be applied from this digital capture (Figure 4); this took another 10 min. The entire data collection and information management process to make it available in CAD was 60 min.

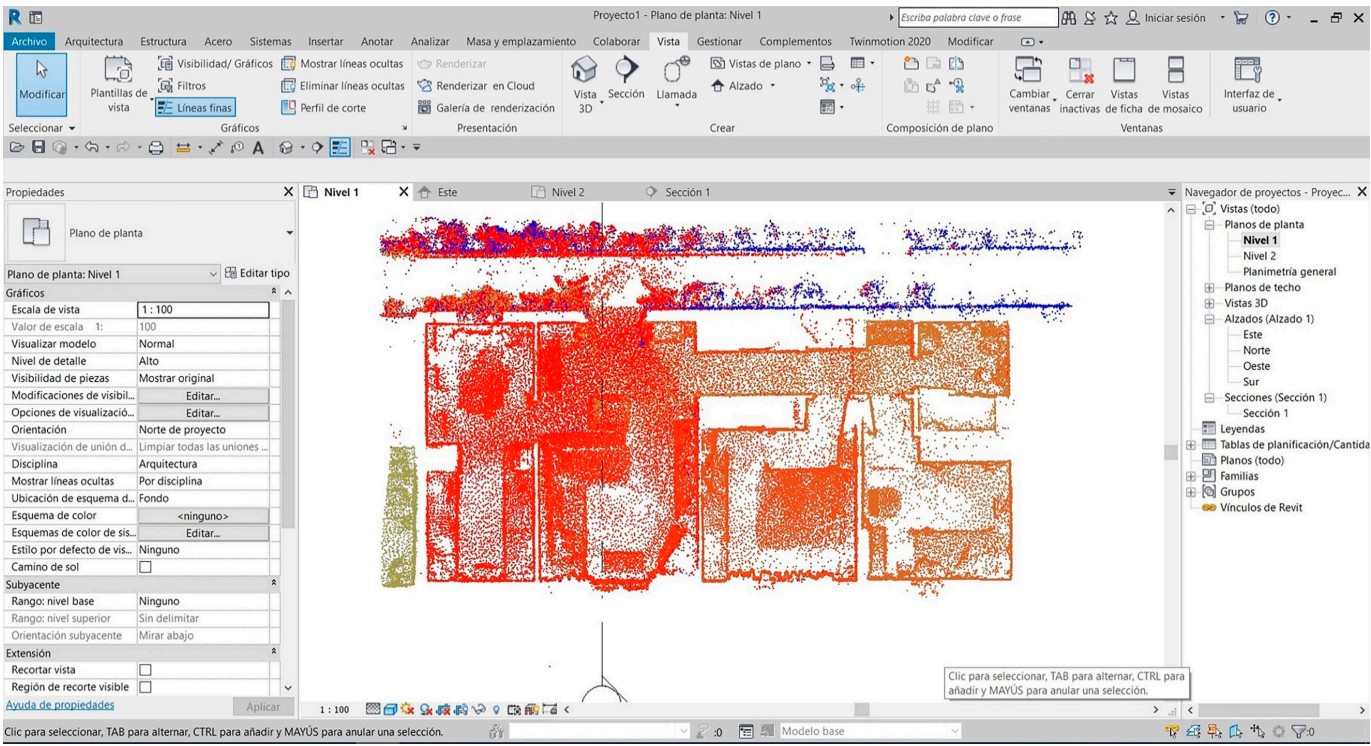

**Figure 4.** Model in Autodesk Revit with the imported point cloud.

A structured methodological process was designed according to the following sequence: analysis of the content and structure of the information of the AVRA program, development of the BIM model, and design of the input and output algorithms between the BIM model and the AVRA program.

### 2.4. Analysis of the Content and Structure of the Information in the Management Software

Information from CMSS-AVRA was received as data in Excel format. The large number of information fields resulted in the output of four separate files, three of which contained general property data and a fourth specifically oriented to the construction injuries registered in the building. The structure of this information was analyzed to identify which parameters allow a link between the CMSS-AVRA and the BIM model and which of these allow its automation from the said model.

### 2.5. BIM Model Generation

The BIM model was developed with the Autodesk Revit program. For the foundation of the model, CAD planimetric documentation and two three-dimensional scans were employed. However, the generation of the BIM model was not based exclusively on the metric information that could be extracted from the CAD vector graphics or from the point clouds of the three-dimensional scan, but rather required a constructive interpretation of the building. This interpretation was based on information obtained from the CMSS-AVRA and from direct inspection of the building itself (Figure 5).

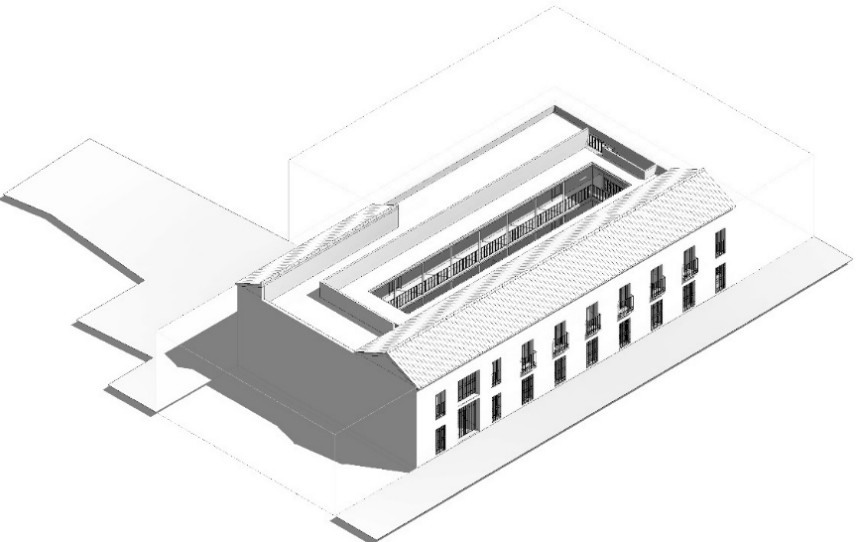

**Figure 5.** General 3D view of the BIM model.

*2.6. Connection between Management Software and BIM Model*

Once the BIM model had been formed and the information fields to be linked between the AVRA application and the model defined, the algorithms that make the transfer between them possible were then programmed. To this end, the Dynamo application, integrated in Revit, was used. An input algorithm was programmed from Excel, obtained from the AVRA application, and another output algorithm from the BIM Revit model to Excel: a format from which the AVRA application could input the information.

## 3. Results

*3.1. Graphical Documentation Strategy*

The implementation of the BIM methodology for the management of large property assets of institutions such as AVRA requires suitable systems and technological resources to render processes viable and sustainable over time. One of the first problems when building these models is the lack of a proper initial graphic documentation since most of these buildings were designed in a predigital era.

The starting point for the development of a BIM model lies in knowledge of the geometry of the building [34–36]. This can be obtained from a metric capture made expressly for the purpose or through an existing previous graphic. Nevertheless, on many occasions this documentation may not exist, may not be found, or does not accurately reflect the current state of the property. Four starting points were established:

a. There is an original prior planimetry, associated with administrative files, that is, hand drawings that were common between the end of the 19th century and the first half of the 20th century, usually in the form of floor plans and elevations, and occasionally as cross sections.

b. There is a description of the property as part of the architectural project or of the final state of the work, reproduced on paper, whose origin is an analogue drawing, common between 1950 and 1990.

c. There is previous digital vector documentation, generally in CAD, that has subsequently been put on paper for the management of the archive, generally from the 1990s onwards.

d. There is no graphic documentation, as they are very old files, or that it is a rehabilitation of a historical building (generally, up to the 19th century), which seldom retains systematic planimetry.

Given these situations, the development of a BIM model raises the choice between a metric capture made expressly for the purpose or the use of existing previous graphic documents. This is an important decision, since it can lead to an investment of considerable time and cost. The decisive factor is the purpose of the BIM model. If the building shows deformations and serious structural problems, or a major reformation is to be carried out thereon, then a comprehensive survey of its geometry would be necessary to develop a rigorous model in which to strictly quantify aspects that cannot be evaluated at first glance, such as overhangs. With the same starting point, if merely the daily preventive maintenance and registration of a building under normal conditions are envisaged, then not only can the model be much simpler, but the starting documentation can also be less metrically demanding. On the other hand, on many occasions, it is necessary to model the building in a simplified and sometimes repeated way for the development of computing applications for calculation, for example, for thermal efficiency.

Therefore, it is necessary to consider both questions, for which we start from Table 1, where the initial graphic information for the construction of the model is proposed according to its purpose.

**Table 1.** Relationship between geometric information support and model requirements.

| | | Geometric Information Needed | | | |
|---|---|---|---|---|---|
| | | **Previous Planimetry (a)** | **Basic Data Collection (b)** | **Comprehensive Data Collection (c)** | **Not Necessary** |
| | Record | | | | ✓ |
| | Day-to-day management/ Preventive Conservation | ✓ | | | |
| Objectives of the model | Specific Reforms | ✓ | ✓ | | |
| | Energy Rehabilitation | ✓ | ✓ | | |
| | Structural Rehabilitation | | | ✓ | |
| | Integral Rehabilitation | | | ✓ | |

(a) Analog or digital; (b) sketch, measurement with conventional techniques; (c) digital capture/laser scanner.

For this research, both options were developed. On the one hand, the existing 2D CAD survey has been used for the generation of the BIM model. On the other hand, an alternative digital capture has been performed using both a manual scanner and a static scanner.

The information obtained from the scan was converted to rcp format, thereby obtaining point clouds that were linked to the BIM model. The comparison between point clouds is one more layer of information that enables the deformations of the elaborated model to be visually verified, and, if necessary, to be modified to better approximate the actual state of the building. In this case, the deformations occurred on the floor plan, due to the adjustment in the execution of the construction to the actual site, but these minor adjustments only affected the minor width of the floor and were insufficiently important as to modify the model (Figure 6).

From this analysis it can be deduced that in those cases where there is not a 2D digital model (CAD type), the best option for the realization of the generic BIM model for the management of the property would be to have a quick data collection of the property through the handheld scanner, which we have verified to be fast and accurate enough in most cases to subsequently obtain a model that is geometrically adjusted to reality. This type of instrument also makes it possible to accompany routine records carried out by technicians to verify any incident, since the transport of the instruments is light and the information management processes are fast.

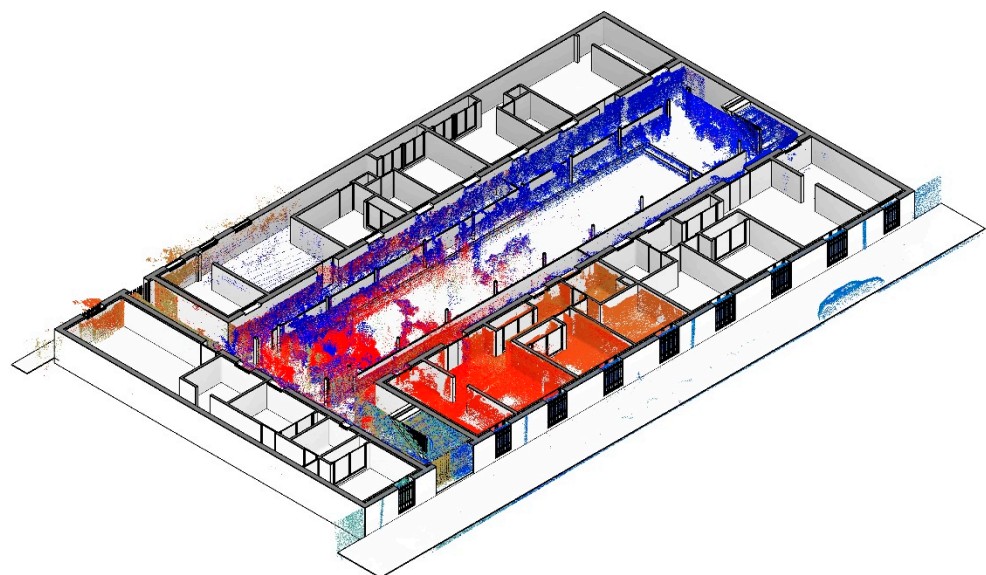

**Figure 6.** Cloud point and BIM model comparison.

Finally, the use of a ground scanner requires a greater investment in time and storage space, although its level of precision is much higher. Solutions to quickly record the data of various static captures already exist today, but typically produce a volume of information that can become unmanageable. For this reason, this type of metric capture is recommended for the analysis of structural problems, or the analysis of persistent pathologies, and should be avoided for problems that require a more generic definition of the model.

The use of the initial documentation requires, in turn, that it be located and linked to the database that contains not only all the records of the assets, but also the basic data regarding the surface area, location, etc. This provides a history to which future BIM models will be subsequently incorporated. The use of this entire graphic legacy has many other documentary functions that can be exploited in the long term, that concern the direct relationship between the forms of representation and the type of architecture developed [37].

### 3.2. Information Structure Analysis

As indicated previously, AVRA currently has software, CMMS-AVRA, for the management of its social housing. According to those responsible for this management, its effectiveness as a management program for the maintenance of its buildings has been sufficiently proven, at least in actions directed by the central services of the agency.

Consequently, the structure and content of the CMSS-AVRA database have been considered as the basis for the information structure to be included in the BIM model. For this, the information contained in the current CMMS-AVRA database was analyzed to determine how the use of BIM models could optimize the current action protocols. In the present investigation, the data related to the case study have been output from the CMSS-AVRA program in Excel format, since this facilitates the bidirectional transfer options with the BIM model. These data include the parameters of different formats and architectural significance, the relevance and usefulness of which in the model required their analysis and assessment. For this, two groups of data have been identified:

The first group includes the data that can be filled in indistinctly in the CMSS-AVRA or in the BIM model, that is, that in which none of the applications offers a significant advantage beyond the opportunity that some of them were already present. For example, the identification of the property will generally be completed in the AVRA program and could be output when a BIM model of one of them is generated. However, in the opposite direction, the foundation of a new BIM model on a property not yet registered in the AVRA application would allow the same identification data to be output (Table 2).

**Table 2.** Informative data that can be entered in both the CMSS-AVRA and the BIM model.

| | | | |
|---|---|---|---|
| **P_CODPRINEX** | 1099 | **P_BARRIO** | Trinidad |
| **P_MATRICULA** | 2070 | **P_TIPOLOGIA** | P |
| **P_DENOMINACION** | MA-0994 | **P_CALLE** | LEMUS |
| **P_NUMVIVIENDAS** | 64-(MA-85/18-AS)/10 VPP | **P_CODPOSTAL** | 29009 |
| **P_PROVINCIA** | 10 | **P_REGIMEN** | ALQUILER |
| **P_MUNICIPIO** | MÁLAGA | **P_AÑOCONSTRUCC** | 1987 |

The second group includes the data that can be extracted directly from the BIM model. These data provide the greatest potential for improvement among the social housing management processes. These data are related to architectural characterization (constructive, structural, functional, etc.) and its quantification. Being architectural information, these data are from the BIM model and, therefore, will have priority over those entered into the CMSS-AVRA. In general, they are quantitative data directly calculated by the BIM software from the modelled elements. For example, the constructed area of the property is an item of data that results from the BIM model and, therefore, is a graphically verifiable item of data that can be automatically updated if an error is detected. However, qualitative data derived from quantifiable data can also be defined. For example, the dominant joinery material is an item of qualitative data determined from the joinery surfaces of each material in the building (Table 3).

**Table 3.** Extract of quantitative data inherent to the BIM model, available to be exported to CMSS-AVRA.

| | | | |
|---|---|---|---|
| **E_SUP_PARCELA** | 542.5 | **CV_OF_DA_SUPERFICIE** | 722.15 |
| **E_SUP_CONSTRUIDA** | 972.75 | **CV_OF_DA_PORCENTAJE** | 72.45 |
| **E_ALTURA_RAS** | 8.10 | **CV_CV_DA_SUPERFICIE** | 106.82 |
| **CV_FP_DA_SUPERFICIE** | 167.75 | **CV_CV_DA_PORCENTAJE** | 10.72 |
| **CV_FP_DA_PORCENTAJE** | 16.83 | **CV_AZ_DA_SUPERFICIE** | 107.36 |
| **CV_FP_AR_PORCENTAJE** | 16.83 | **CV_AZ_DA_PORCENTAJE** | 29.6 |

From what is revealed in the analysis of the content and the information structure of the CMSS-AVRA, it can be inferred that the proposed BIM model must not be a mirror reflection thereof, but rather that the optimization of the information is proposed in terms of reliability, auditability, and automation: reliability in terms of the minimization of manual data transcription errors; auditability since the graphic condition of the BIM database allows both the visual recognition of the origin of the data in the three-dimensional model and the confirmation of its veracity; and automation in relation to the direct update of the data after the eventual modifications of the elements of the model. These specific advantages of employing BIM in this case study complement other advantages that are inherent in its nature as a graphical database and that are also useful in this context: the three-dimensional characterization and the availability of coherent planimetric documentation (Figure 7).

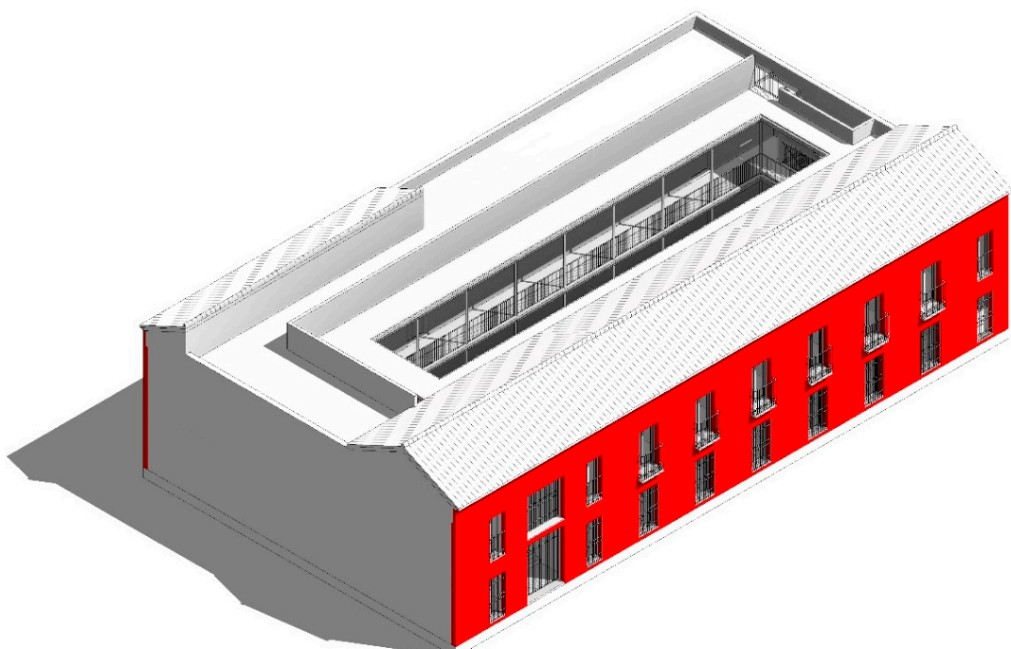

**Figure 7.** Data auditability. The parameter CV_FP_DA_SUPERFICIE filtered by color, which shows the area of the exterior façades.

### 3.3. Generation and Characterization of the BIM Model

The generation of the model involves both the geometric 3D modelling of the building and the design of the linked information structure, described in the previous sections. Geometric 3D modelling is based on a digital capture or on existing graphic documentation. In this case, both options have been explored to build the model, although the existing CAD survey has been sufficient to produce an operational model in accordance with AVRA's management requirements. Similarly, a linked information structure has been designed. The connection between the graphic elements and their information is solved thanks to the architectural analysis, which makes it possible to determine which information corresponds to each element of the model.

The resulting model is called BIM-AVRA, suitable for attending the management of a social housing manager. Its main characteristic is to be a 'live' model, i.e., a model that is systematically updated. To ensure the genuine effectiveness of this systematic update, the BIM-AVRA model must be as simple as necessary to ensure that the manager, in our case, AVRA, can bear the costs involved in the update process. In any case, the simplification must guarantee that the model includes the basic configuration of the building, but no elements that may be altered by third parties, whether tenants or other types of users. For example, the BIM-AVRA model will not include the color of an interior wall or the form of the bathroom cladding because these elements may be altered by third parties without the knowledge of the manager, thus rendering the model out of date (Figure 8).

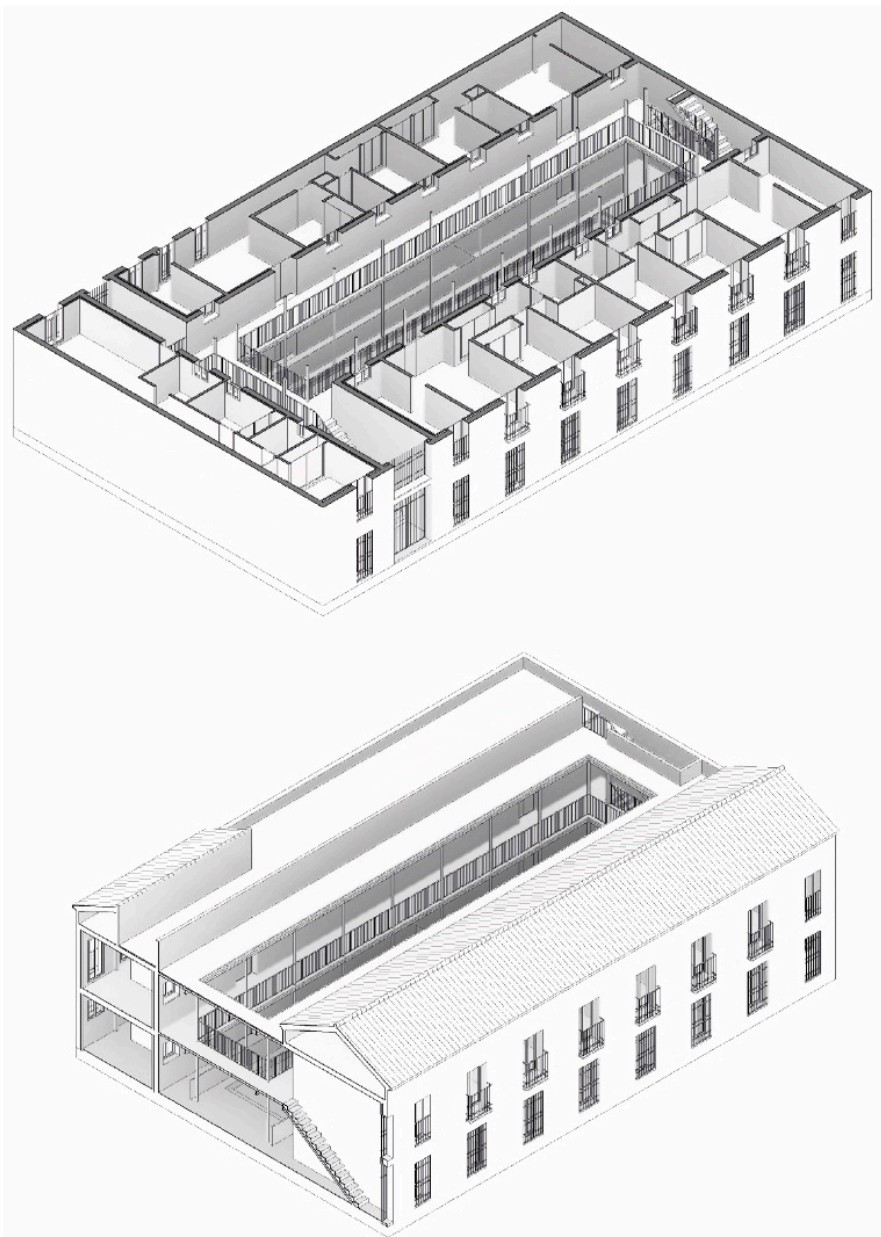

**Figure 8.** Horizontal and vertical section 3D views of the BIM-AVRA model of the case study.

Therefore, this model differs from others that could be created for specific new-build or refurbishment interventions. In other words, a single model cannot resolve the entire complexity of the useful life of a building, although it is possible to successfully integrate different models generated during its useful life. The BIM-AVRA model can offer basic architectural information about any of the agency's buildings and, therefore, can become the base model for any other models that may be generated for specific events during the useful life of a building, such as its extension, refurbishment, or demolition. These models would therefore serve a very different purpose, with the specific aim of documenting a design, execution, or state following an execution. Keeping these models separate from the BIM-AVRA model mitigates the problems derived from outsourcing the design and execution of a building because the models developed for these purposes will be independent from the base model and may be as detailed as required by the specific intervention addressed. Situations such as nonexecuted designs, partial studies, and building analyses, or even information regarded as an inaccurate reflection of the state of the existing building but useful to keep, will be archived as "still photographs" of the

intervention without compromising the rigorous updating of the BIM-AVRA model. When these intervention models become substantial alterations of the building, they will be incorporated into the updated model in a simplified format consistent with the existing data in the model (Figure 9).

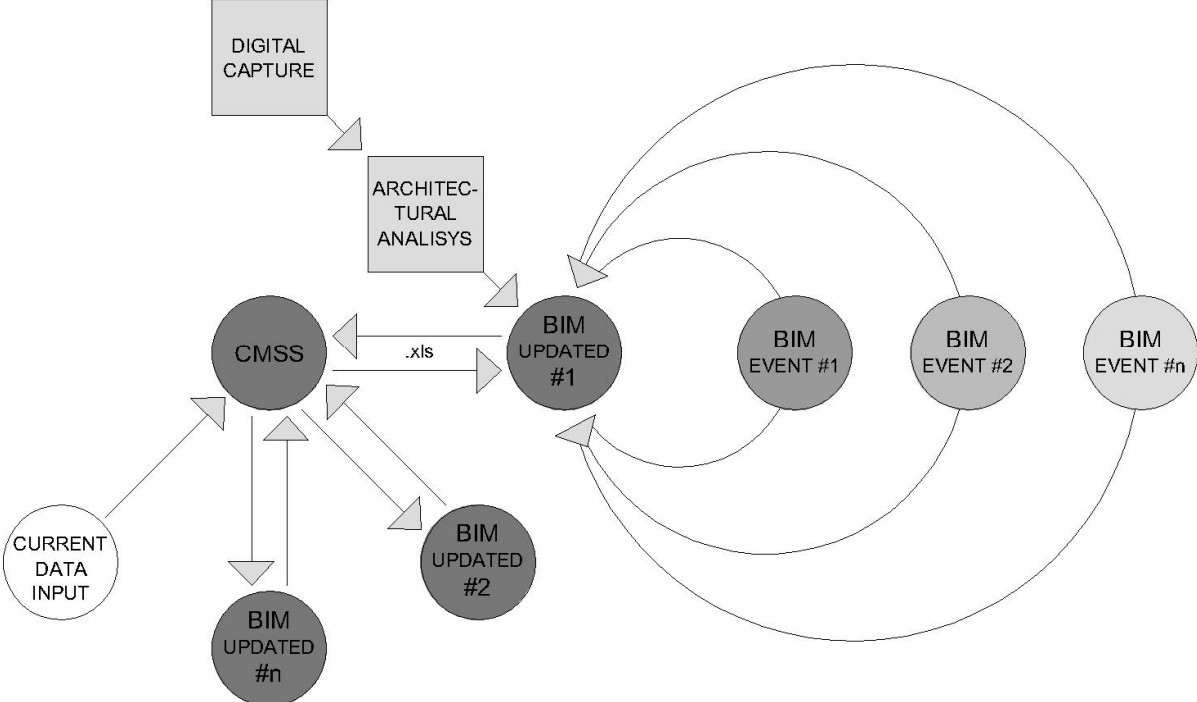

**Figure 9.** Diagram of the life cycle workflow around the updated BIM model.

### 3.4. Data Transfer and Interoperability BIM-AVRA/CMMS-AVRA

The algorithms that allow the transfer between the BIM model and the AVRA program are programmed with Dynamo, an application integrated in Revit that allows Excel files to be used as a format for the input and output of the model information. Both processes are structured by 'categories', that is, the set of objects that within a BIM model are associated with a certain constructive function (walls, roofs, doors, windows, etc.) or characterization of the model (information, project, built areas, etc.).

In the case of input from Excel, the Dynamo algorithm starts reading the selected data to assign them to the corresponding categories in the model. The process in the opposite direction is symmetric, although one must stop at articulating the data lists in a way that is consistent with the sequence expected by the AVRA management program (Figures 10 and 11).

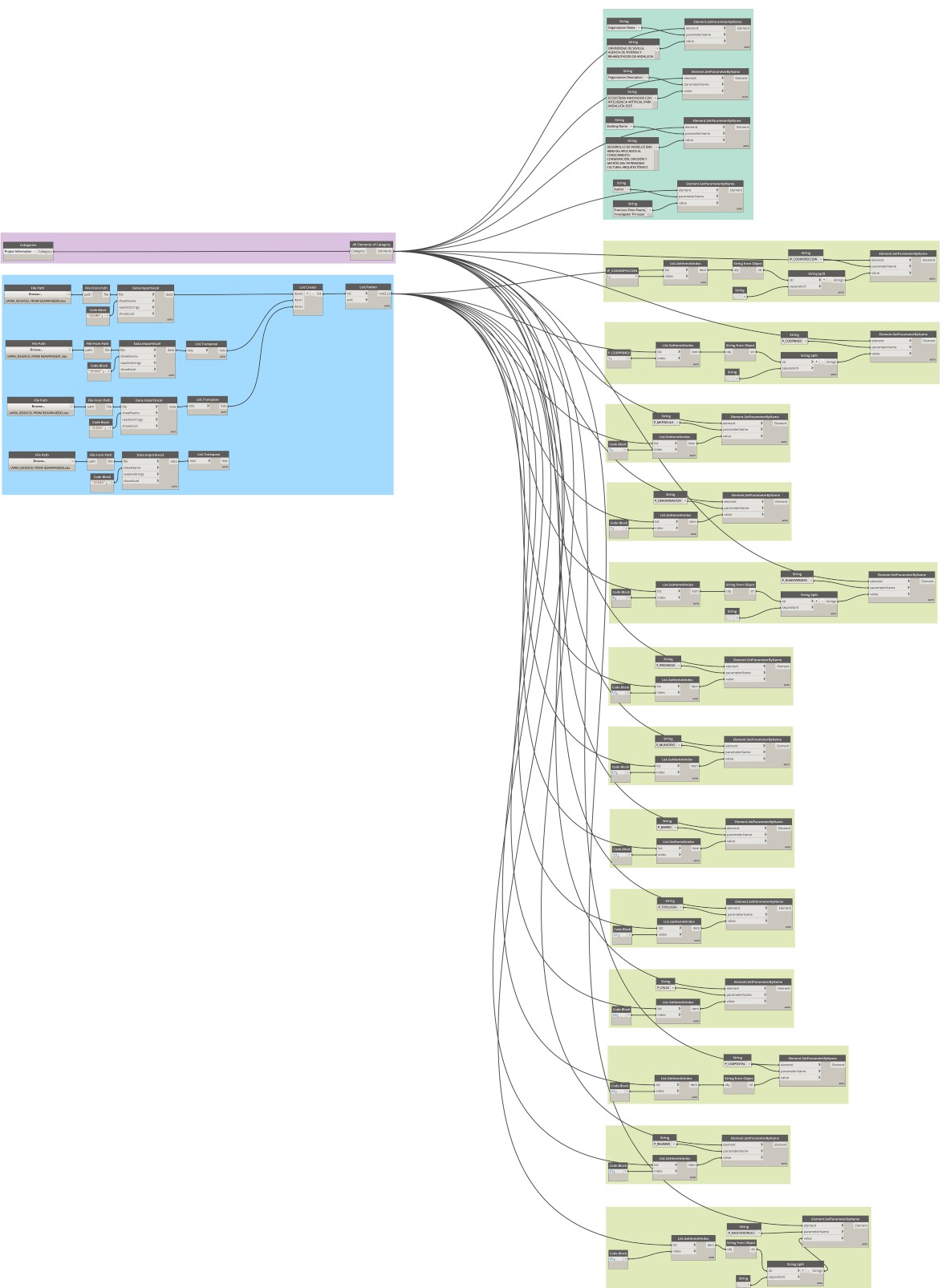

**Figure 10.** Dynamo algorithms for data output from Excel to the BIM-AVRA model (IMPORT.dyn).

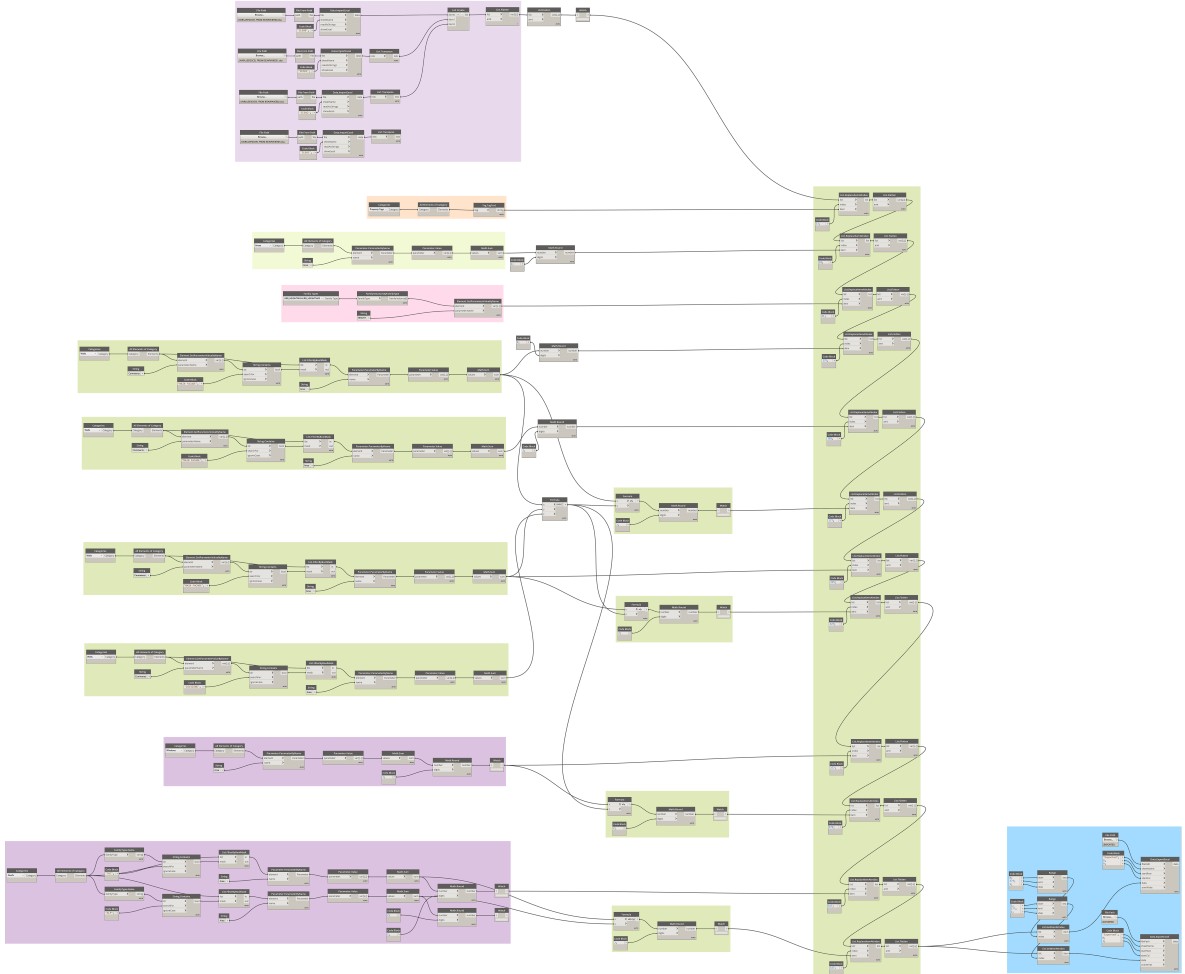

**Figure 11.** Dynamo algorithms for data output from the BIM-AVRA model to Excel (EXPORT.dyn).

### 3.5. Summary of the Proposed Workflow

The proposed workflow is organised around two main cores: the CMMS software of the managing institution, in our case CMMS-AVRA, and the updated BIM model of the building, in our case BIM-AVRA (Figure 9). The CMMS is a single core, and the BIM model is a multiple core, so there will be as many updated BIM models as there are buildings in the building stock to be managed. This does not mean that the workflow is only operational when all updated BIM models have been generated, but it can be combined with the current workflow. CMMS management can remain operational with the current data input system, which, as described in the Introduction, is mainly manual and, as updated BIM models of the buildings are generated, the system becomes more reliable, auditable, and automated.

Once the action on a building has been decided, the process of generating the updated BIM model begins. For this, existing graphic documentation or, where appropriate, digital capture will be used, in accordance with the strategies described in the previous points. Regardless of the sources of graphic documentation for the generation of the model, it is essential to perform an architectural analysis to basically characterise the structural system, the construction system, and the MEP system. This analysis allows the level of development of the model to be adapted to the knowledge available and to the resources available to the manager to keep it updated.

During the life cycle of the building, a number of unique events will occur, depending on the different types of intervention that the building manager may decide to carry out on the building. The updated BIM model provides reliable information for the intervention project and can serve as a basis for the generation of a BIM model specifically oriented to the objectives of that intervention. In other words, a BIM model of the corresponding event

will be produced based on the updated BIM model, but different from it. Once the event is completed, the team responsible for maintaining the updated BIM model will revert the changes to the BIM model, according to its own standards and level of development. The BIM model used to produce the intervention will be archived and document the event at that point in the building life cycle.

At any time, the bidirectional communication of the updated BIM model with the CMMS can be updated via the xls export and import process described in the previous sections.

## 4. Discussion and Conclusions

This research presents a workflow to improve social housing life cycle management using BIM. To this end, the AVRA case study has been considered, given its role as a public owner that manages more than 75,000 dwellings. Based on unstructured interviews with those responsible for AVRA's property management, the following problems were stated: alphanumeric and graphic documentation and its use as an effective decision-making assistant for real estate.

The first step for BIM-based social housing management is the definition of a strategy for the graphic documentation of the buildings to be managed. The generation of the BIM model can be approached: (a) from 2D digitized plans (dwg, dxf or similar); (b) from plans on hardcopy; and/or (c) from the building itself by means of a digital and/or photogrammetric capture. Although digital capture offers advantages such as speed and accuracy, it is also a high-cost investment when it comes to hundreds of surveys. In the case study, two options were tested: the generation of the model from 2D CAD plans and a digital capture. It was concluded that the key point in making a choice is the purpose of the model, and a table was presented with the different purposes and recommended resources. For the day-to-day management of such a number of buildings as AVRA manages, the generation of the BIM model from 2D CAD drawings was proven to meet the proposed objectives.

On the other hand, digital capture by scanner was shown to be an effective and efficient procedure for generating BIM models. Two alternatives were studied for the metric capture of buildings: I. using a manual scanner; and II. a static scanner. It has been verified that, whatever the case may be, the information taken is compatible and can be related to the final BIM model, whether it is formed based on the digital capture carried out or not. The use of a manual scanner presents the advantages of transportable equipment, speed, simplicity, and efficiency of the data capture procedure compared to the static scanner, which requires more time for preparation and data collection. However, the static scanner presents greater precision. Given these results, it can be concluded that the point cloud obtained with the handheld scanner offers suitable precision for the realization of a general building management model. The point cloud obtained by means of the static scanner, more precise than the previous scanner, is useful in buildings that require intervention from a constructive and/or structural point of view.

The key parameters of the BIM model were determined to enhance the management carried out together with the CMSS-AVRA. This means that the information structure of the model is not a mirror of the CMSS-AVRA information, but the definition of two groups of data: those that are simply informative and those that quantify the architectural characteristics of the building. Informative data can be added to both software, but the source of quantitative architectural data must be the BIM model, as this ensures reliability, auditability, and automation.

The characterization of the BIM model oriented towards the sustainable management of the building's life cycle has been determined. In the modelling strategy, the requirement for simplicity prevails, modulated by the manager's capacity to carry out up-to-date maintenance that guarantees its long-term operability. Although simplified, the model must incorporate the basic configuration of the building, leaving out everything that can be altered by third parties. Consequently, it is ruled out that a single model can assume the full complexity of the building's life cycle, although it is possible to effectively

integrate different models generated throughout the building's life cycle. A workflow in which interaction with the CMSS-AVRA is always carried out from this updated BIM-AVRA model is proposed. This updated BIM-AVRA model is offered as a basis for the development of other specific models for different events in the life of the building. In turn, the BIM models for each event will be archived and only the information relevant for the day-to-day maintenance of the building will be transferred to the BIM-AVRA model.

The connection between CMSS-AVRA and the BIM model has been solved, allowing the flow of information between both. A Dynamo script was programmed to automatically transfer the selected CMSS-AVRA data to the BIM model (IMPORT.dyn) via an Excel file. In turn, a Dynamo script was also programmed to automatically transfer the data contained in the BIM model to CMSS-AVRA via an Excel file (EXPORT.dyn). The latter, obtained directly from the BIM model, improves the management workflow, as they are reliable, auditable, and automated.

Finally, AVRA technicians have been shown the joint use of the CMSS-AVRA and the BIM-AVRA model of the case study, expressing their interest in the future developments of the results of this research. A first line of development would be oriented towards the extension of this experience to a sufficiently significant number of models to produce strategic decision-making assistance programmes. This would require selecting the key parameters of BIM models and combining them with others specifically designed to represent the strategic criteria of managers. A second line of development would be oriented towards the characterization of the modelling of singular buildings. In this sense, it would be particularly interesting to define specific strategies for the modelling of heritage architecture, incorporating artistic, historical, and archaeological information. Heritage applications of BIM, known as HBIM, have been widely tested in recent years, but it would be very interesting to reconcile their contributions with the management requirements of social housing use.

**Author Contributions:** Conceptualization, A.G.-M. and M.C.-R.; methodology, A.G.-M. and M.C.-R.; BIM model and Dynamo scripts, M.C.-R.; validation, M.C.-R., A.G.-M., M.L.P.L.; formal analysis, M.C.-R. and A.G.-M.; investigation, M.C.-R. and A.G.-M.; data curation, M.C.-R.; writing—original draft preparation, M.C.-R. and A.G.-M.; writing—review and editing, A.G.-M. and M.C.-R.; visualization, M.C.-R.; supervision, A.G.-M. All authors have read and agreed to the published version of the manuscript.

**Funding:** This research has been financed by European Regional Development Funds and the Junta de Andalucía, through the CEI-10-HUM799 project, within the framework of singular projects of transfer actions in the Campus of Excellence in the areas of the Research Strategy and Innovation (RIS3), within the global project of the University of Seville Innovative Ecosystem with Artificial Intelligence for Andalusia 2025.

**Institutional Review Board Statement:** Not applicable.

**Informed Consent Statement:** Not applicable.

**Data Availability Statement:** Not applicable.

**Acknowledgments:** The authors thank the Andalusian Housing and Rehabilitation Agency AVRA, the Ministry of Development, Infrastructures and Spatial Planning, and the Junta de Andalucía for providing all the information necessary for the development of the case study proposed in this investigation. We are especially grateful for the work carried out by Isabel Ramírez Acedo and Miguel Ángel Lobato Aguirre; the technical area members, Elena Morón, Mª María Bermejo Oroz, Jorge Ruiz García; the AVRA work team of the delegation of Málaga, Miguel Ángel Santos Amaya (Head of the Development and Quality Control Service, Ministry of Development, Infrastructure and Land Management—SGT); and Francisco Javier Ramos from the company Geoavance SL for his availability and kindness in accompanying and supporting the data collection. We also appreciate the contribution of the research project "Development of BIM-HBIM-GIS models applied to the knowledge, conservation, dissemination and management of architectural cultural heritage", especially the kind support of its coordinator Francisco Pinto Puerto and the researcher José María Guerrero Vega.

**Conflicts of Interest:** The authors declare no conflict of interest. The funders had no role in the design of the study; in the collection, analyses, or interpretation of data; in the writing of the manuscript, or in the decision to publish the results.

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
