# Peer review of "Social Housing Life Cycle Management: Workflow for the Enhancement of Digital Management Based on Building Information Modelling (BIM)"

_sustainability, doi:10.3390/su14127488_

Round 1

Reviewer 1 Report

Dear authors

I congratulate you for the idea and the writing of the article.
To have more benefit I signal you some notes:
1. attention to editing. There are several typos (dots, marks, translation of specific words: e.g. what is a "clericla work"?)
2. the work explains in detail some phases that are perhaps quite obvious, such as the scan to BIM and the rendering on a specific autoring program [REVIT by Autodesk]. In these cases it is recommended to propose as research the potential of BIM in a broader way, but especially to highlight the innovative steps and the specific algorithms written to achieve the proposed purpose.
3. little space is dedicated to the specifics of the workflow developed for AVRA, there are no tables, nor work chronoprograms explaining the proposal and comparing the advantages: It is convenient to insert them.
4. in general, references to examples of real estate management are also poor, some of the results considered innovations seem to already exist in the state of the art of real estate management and the novelty is not grasped. it would be worthwhile to make this clearer and go into greater detail.

The topic is proposed as an article.
It would appear to be more of a case study.

Author Response

Please find the cover letter attached below.

Reviewer 2 Report

Thank you for your contribution.
The integration of BIM in the management of huge building stocks (such as social housing) is a solid research theme, and the exploration of this integration is encouraging and promising, as it could result in strong benefits to all the stakeholders engaged in the process.
The authors aim to define a workflow to facilitate this integration.

Nonetheless, the paper appears to be premature and discursive. The structure is uncertain and unclear; for instance, lines 98-113 are related to the methodology, but the “Methods” section is referred to the case study, and lacks the proper methodology of the research. The section dedicated to results appears very digressive: the effectiveness of the proposed methodology is not measured through KPIs or else.

The proper goal expressed in the title and in the abstract refers to the definition of a workflow, but no processes or flows can be found in the manuscript. The analysis of existing workflows is discursive: no representation or detailed description of this workflow is provided. Also, the proposed BIM-based workflow is not described in any form.
A general lack of detail can be highlighted, with the exception of the part dedicated to the description of the geometrical survey. BIM model requirements and levels of details depending on use is a vast and interesting part of BIM research, but no detail is provided on the information requirements for buildings life cycle management.

There are many bullet or numbered lists that make the text fragmented and unclear (e.g., lines 60-76, 122-131, 136-138, 147-151). I suggest describing unambiguously the goals and the methodology used to reach those goals.

I believe that the manuscript would benefit of a large rework to better define the goal of the research, the methodology proposed, the application on a case study (that should be a validation of the methodology), and the evaluation of measurable results provided.

Author Response

PLease find the cover letter attached below.

Round 2

Reviewer 1 Report

Dear authors,

overall the contribution has certainly improved and we congratulate you on your work.
Just a few suggestions for an accomplished closure:
- the discussions lack references to the digitial management systems in use in some Europe, in the USA and also in experimentation in Asia. it is important, even if only briefly, to place the proposed experience among other significative cases.
- the conclusions should not be limited to the possibility of the proprietary administration to apply the workflow, but to its openness to future research.

Reviewer 2 Report

I appreciate the improvements carried out on the manuscript. I provide some observations:

  • It is not clear what kind of information you are referring to in the manuscript, as the proper maintenance or management activities are not described. The software used for building management (AVRA) is never described.
  • The references are limited and do not provide a proper overlook of the research theme.
  • A detailed description of the methodology is still lacking, as the paper is centered on the case study. For instance, if the objective of the paper is a workflow, I suggest to provide a flowchart with the method proposed.
  • It is not clear what kind of information you are referring to in the manuscript, as the proper maintenance or management activities are not described. The software used for building management (AVRA) is not described in terms of information required.
